# Food security in rural South Africa: The role of household head demographics, crowding, and wealth

Reneilwe G. Mashaba[1], Cairo B. Ntimana[1,2]*, Katlego Mothapo[1],
Kurisani M. Mabhedle[1], Joseph Tlouyamma[1], Kagiso P. Seakamela[2]

1 DIMAMO Population Health Research Centre, University of Limpopo, Sovenga St, Polokwane, South Africa, 2 Department of Pathology, University of Limpopo, Sovenga St, Polokwane, South Africa

☙ These authors contributed equally to this work.
* cairo.ntimane@ul.ac.za

## Abstract

Food insecurity, characterized by uncertain or limited access to adequate food, remains a pressing public health issue. South Africa, particularly its rural areas, continues to experience significant levels of food insecurity, exacerbated by economic inequality and structural barriers. This study aimed to investigate how the demographics of the household head, household crowding, and wealth influence household food security in rural Limpopo Province, South Africa. A cross-sectional study was conducted using routinely collected data from the DIMAMO Health and Demographic Surveillance System, including 17,374 household heads in rural Limpopo Province. Data was analyzed using STATA 16.1. Both bivariable and multivariable regression reported that an increase in the household head's age was negatively associated with food insecurity. Similarly, household heads in the middle category of the wealth index were protective of food insecurity (OR=0.73; 95%CI:0.59-0.90). In binary logistic regression, household crowding was associated with food insecurity (OR= 2.03; 95%CI: 1.65-2.49). Similar results were noted in multivariable regression, even after adjusting for possible confounders (AOR=2.62; 95%CI: 1.75-3.90). Divorced household heads were also associated with food insecurity (AOR=5.39; 95%CI:2.88-10.12). The age of the household head was reported to be a protective factor for food insecurity (AOR=0.17;95%CI:0.10-0.31), with food insecurity decreasing as age increased. The present study noted a low prevalence of food insecurity (3.51%). The low prevalence of food insecurity suggests that social protection mechanisms and local indigenous farming are a crucial barrier to households' food scarcity. Household food insecurity in rural Limpopo is influenced by household head age, household crowding, marital status, and wealth, rather than gender, education, or employment status. These findings highlight the need for targeted policy responses, including the extension of income-support mechanisms to unemployed younger household heads, focused social support for widowed households, and

**Data availability statement:** For data access, requests may be directed to the University of Limpopo Research Ethics Committee, which serves as the institutional body responsible for data governance and ethical clearance related to this study. Their contact information is as follows: University of Limpopo Research Ethics Committee Email: ethics@ul.ac.za.

**Funding:** The authors received no specific funding for this work.

**Competing interests:** The authors have declared that no competing interests exist.

community-based interventions aimed at reducing the economic strain associated with household crowding.

## Introduction

Food insecurity, "defined as a situation whereby people lack regular access enough to safe and nutritious food for normal growth, development, and an active healthy life due to financial, physical, or other resource constraints " is a persistent public health challenge [1–3]. The negative implications of food insecurity include poorer health outcomes and an increased predisposition to chronic conditions such as diabetes and high blood pressure [4]. According to the World Health Organization, the prevalence of moderate or severe food insecurity was 29.6 percent in 2022 [5]. Sub-Saharan Africa is highly impacted by this, with an overall prevalence of high food insecurity of 36.4% [6]. In South Africa, the prevalence of food insecurity has been on the rise [7]. According to STATS SA, the proportion of households in South Africa that experienced moderate to severe food insecurity was estimated at 15,8% in 2019, 16,2% in 2022, and 19,7% in 2023 [8]. Over this period, the proportion of households that experienced severe food insecurity was estimated to be 6,4%, 7,5%, and 8,0%, respectively [8].

Household crowding and wealth index have been reported to be associated with food insecurity in the literature [9–11]. The household size and the number of people in the household may strain household resources and limit food availability [12]. Similarly, households with lower wealth index scores are more susceptible to food insecurity due to limited financial resources, which restrict access to adequate and nutritious food [13,14]. These structural factors can increase the effects of other socio-demographic risks and exacerbate food insecurity in resource-limited settings [15].

In South Africa, efforts have been made to reduce food insecurity and improve livelihoods. These include the provision of social grants [16]. However, these grants are generally not sufficient to meet the household dietary needs; hence, rural areas remain overly exposed to food insecurity [17]. For instance, approximately 52% of the rural households in Limpopo Province of South Africa were considered severely food insecure, contributed to by high levels of unemployment in the province [18]. Furthermore, household and household head demographics have been reported to play a role in food security. A systematic review conducted in Ethiopia reported that households headed by females, low educational attainment, and unemployment were associated with food insecurity, while an increase with age, higher education, and high wealth index were protective barriers for food insecurity [19]. Similarly, a study conducted in Nigeria noted similar findings with female-headed households being at risk of food insecurity [20].

Although predictors of food insecurity (poverty, low education, female-headed households, and large household size) have been reported globally, much of this evidence emanates from different socio-economic and cultural factors [21,22]. These factors may not that may not apply to rural South Africa, due to differences in social protection systems, labour migration patterns, and household structures [23,24]. For

example, the role of household head demographics characteristics, household crowding, and wealth indices in influencing food security within rural South African communities remains not fully explored [25–28]. Hence, the present study aimed to associate household food security with the demographics of the household head and the household-level characteristics, such as household crowding and wealth index in Limpopo province, South Africa.

## Methods

### Study design, setting, and population

The study used data routinely collected from the DIMAMO HDSS. The site is located approximately 35 km northeast of Polokwane in Limpopo Province, within the Polokwane Local Municipality of Capricorn District. It is in close proximity to the University of Limpopo (Turfloop campus) and lies between the coordinates 29.65° and 29.85°E and 23.65° and 23.90°S [29]. The area lies within a semi-arid agro-ecological zone with warm summers, mild winters, and seasonal rainfall that falls mainly from October to March. Agricultural activities in this area depend on rainfall, and households rely largely on subsistence crop production and small-scale livestock farming. Consequently, rainfall variability directly influences local food production and household food availability [30]. The region is characterized by low socio-economic status and is predominantly inhabited by individuals of African ancestry, primarily from the Ba-Pedi ethnic group.

This study utilised secondary data from the DIMAMO Population Health Research Centre (PHRC) database, which forms part of the DIMAMO HDSS. The DIMAMO HDSS is a population-based longitudinal surveillance platform that continuously monitors demographic, socioeconomic, and health-related characteristics of households within the surveillance area. Data in this system are collected through routine household visits conducted by trained fieldworkers using standardized structured questionnaires. These surveys capture information on household composition, education, employment status, housing conditions, household assets, and food security indicators, among other variables. The surveillance system operates under the South African Population Research Infrastructure Network (SAPRIN) and follows standardized procedures for data collection, verification, and quality control, making it an appropriate and reliable data source for population-level public health research. More details about the DIMAMO PHRC are cited elsewhere [31–33].

### Data preparation and instruments

Household-level data collected through the DIMAMO HDSS were extracted from the DIMAMO PHRC database on 5 May 2025. At the time of extraction, the surveillance database contained records of households residing within the DIMAMO surveillance area. From this dataset, 17,374 household heads with complete information on food security indicators and key socio-demographic variables were included in the present analysis. Household heads were selected because they are typically responsible for household resource allocation, food acquisition, and household decision-making, making them appropriate respondents for examining determinants of household food security.

The variables extracted from the database included household head demographic characteristics (age, gender, marital status, education level, and employment status), household characteristics (household size and number of bedrooms), household assets used to construct the wealth index, and food security indicators based on 12-month recall questions. DIMAMO PHRC collects the 12-month recall Food Security questions. To best capture both chronic and seasonal variations in food availability, the 12-month recall period was rendered appropriate for assessing household food insecurity in this rural, rain-fed agricultural area [34]. In addition, longer recall periods are commonly used in population-based surveys [34].

### Wealth index construction

To assess the relative socioeconomic status of households, a wealth index was constructed using principal component analysis (PCA). We included a set of household assets and amenities that are commonly used to measure long-term economic status. These included items such as cooking appliances, a refrigerator, a television, a bed, a table and chairs,

bicycles, a car/bakkie, livestock, and access to electricity-dependent appliances. Only binary variables with variation were retained, and variables with little or no variability were excluded. To determine adequate inter-item correlations, a correlation matrix was used. The selection of binary variables was assessed using the Kaiser–Meyer–Olkin (KMO) measure. Bartlett's test of sphericity was performed to confirm that the correlation matrix was not the same. The principal components that were retained were selected based on eigenvalues >1 and inspection of the scree plot. The first principal component, which explained the largest proportion of variance, was used to determine the wealth index score. To facilitate categorical analysis, the wealth index was divided into tertiles. Households were categorized into low, middle, and high socioeconomic status.

## Determination of food insecurity

Researchers constructed a food insecurity score based on 18 binary indicators of food-related hardship, adapted from items conceptually aligned with the Household Food Insecurity Access Scale (HFIAS) [35]. These indicators included experiences such as reducing meal size, skipping meals, and running out of food, as well as their reported frequencies. Each experience was counted once if it occurred, resulting in an additive score ranging from 0 to 18. The total score was then grouped into four categories to reflect increasing severity of food insecurity, similar to the HFIAS categorization approach: Food secure (score = 0), Mildly food insecure (score = 1–2), Moderately food insecure (score = 3–7), Severely food insecure (score = 8–18). For analysis, the four food security categories were converted to a binary variable (food secure = 0, food insecure = 1). We adapted the food insecurity indicators from HFIAS because the dataset did not contain the original HFIAS questions. These items mapped closely to the frequency-based HFIAS domains, allowing us to construct a comparable measure. Although the adapted scale is not the full HFIAS, similar adaptations have been used in household surveys where only partial or modified items are available [36–38].

## Determination of household crowding

The crowding ratio was defined as the number of household members divided by the number of bedrooms within the household [39,40]. The crowding ratio was then divided into tertiles to reflect varying degrees of household density: low crowding (1st tertile), medium crowding (2nd tertile), and high crowding (3rd tertile).

## Statistical analysis

Data analysis was conducted in STATA 16.1 (Stata Corporation, College Station, Texas, USA). Continuous variables that are normally distributed were summarized using mean and standard deviation, while variables that were not normally distributed were summarized using median and interquartile range. Categorical variables were represented as percentages, and a chi-square test was used to assess the association between variables. Multivariable logistic regression models were used to describe the relationship between the binary dependent variable (Food secure = 0, Food insecure = 1) and the predictor variables. Crude odds ratios were estimated by including only the dependent and one predictor variable. Model checking was performed using a Variance Inflation Factor (VIF) statistical test on unweighted data. VIF < 5: Low multicollinearity. VIF between 5 and 10: Moderate multicollinearity. VIF > 10: High multicollinearity, suggesting potential issues [20]. All variables had VIF values below the commonly used threshold of 10, indicating no problematic multicollinearity. The crude and adjusted odds ratios were reported with 95% CIs, and p-values of less than 0.05 were considered statistically significant.

## Ethical considerations

The study was conducted in accordance with the Declaration of Helsinki. This study has been approved by the University of Limpopo Research Ethics Committee (TREC) (TREC/07/2025: IR).

## Results

The present study included 17374 household heads. The overall prevalence of food insecurity was 3.51%. Male-headed households represented 47.17% while female-headed households represented 52.83%. There was no significant difference in food security level concerning the gender of the household head [Food insecurity (male = 3.28%, and female = 3.70%)] with a p-value of 0.131. Most household heads (63.72%) were in the age category 40 – 69 Years old. Food insecurity decreased with age [< 39 Years old (7.11%), 40 – 69 Years old (3.59%), and ≥ 70 Years old (2.20%)] with the P value of < 0.001. There was a significant association between marital status and food security (p<0.001). Widowed and divorced individuals have higher rates of food insecurity (8.12% and 4.29%, respectively) compared to those who are single or married. The present study found no significant difference in food security status by the household head's educational and employment status. However, most household heads had secondary education as their highest level of education (73.97%), with about 73.36% unemployed. Household crowding played a significant role in food insecurity. Households with high crowding had the highest level of food insecurity (p-value <0.001). There was a significant difference (p-value = <0.001) in food security by household wealth index (poor: 3.69%, middle: 2.73%, and rich: 4.09%) (Table 1).

In bivariable logistic regression, households headed by younger individuals were likely to be food insecure. Household heads who were aged 40 – 69 years old were less likely to experience food insecurity (COR = 0.49; 95% CI:0.39- 0.61; p<0.001). The likelihood of experiencing food insecurity was even lower in the ≥ 70 years old category (COR = 0.30; 95% CI: 0.03-0.39; p<0.001) compared to the other age categories. Household heads who were married (OR =1.34; 95% CI: 1.24-1.91, *p*<0.001) and widowed (COR = 3.04; 95% CI: 1.77- 4.23; p<0.001) were reported to be associated with food insecurity. Household head's gender, educational status, and employment status were not associated with food insecurity. Household heads in the middle category of the wealth index were protective of food insecurity. Households with medium levels of crowding had significantly higher odds of being food insecure (COR = 1.39; 95% CI: 1.12–1.72, *p*=0.003), while those with high levels of crowding had even greater odds (COR = 2.03; 95% CI: 1.65–2.49, *p*<0.001) (S1 Fig and S1 Table).

The age categories 40 – 69 Years old and ≥ 70 years were reported to be protective factors for food insecurity. Household head's gender, educational status, and employment status were not associated with food insecurity. Divorced household heads were associated with food insecurity (AOR = 5.39; 95%CI:2.88-10.12; P<0.001). Medium and high household crowding were 2.03 and 2.62 times more likely to be associated with food insecurity, respectively. The odds of food insecurity were reduced by 50% among study participants from the rich wealth category compared with those from the "poor" wealth category. (p<0.001) (S2 Fig and S1 Table).

## Discussion

The present study aimed to associate household food security with the demographics of the household head and household-level characteristics such as household crowding and wealth index in Limpopo province, South Africa. The study found a low overall prevalence of food insecurity (3.51%). An increase with age of the household head was noted to be a protective barrier against food insecurity; on the contrary, household crowding was associated with higher odds of food insecurity. Food insecurity decreased with an increase in the household wealth index. Widowed participants were associated with increased food insecurity. There was no significant association between household head gender, education, and employment status with food insecurity.

The present study included 17374 household heads (47.17% male-headed and 52.83% female-headed). The overall prevalence of food insecurity was 3.51%. The prevalence reported in the present study was lower than the national prevalence of food insecurity in South Africa, which was reported as 23·7% [41]. This is considering that other studies conducted in other parts of Limpopo reported food insecurity prevalence of between 11.4% and 44.4% [42,43]. Although rural areas are commonly reported to have a high prevalence of food insecurity, factors such as subsistence farming

PLOS Global Public Health

**Table 1. Baseline distribution of household food security status by household head and household characteristics.**

| Variables | Overall | | Food secure N (%) 16764 (96.49%) | | Food Insecure N (%) 610 (3.51%) | | |
|---|---|---|---|---|---|---|---|
| | N (%) | 95% CI | N (%) | 95% CI | N (%) | 95% CI | P-value |
| **Household Head Age** | | | | | | | |
| < 39 Years old | 1490 (8.58) | 8.0, 9.0 | 1384 (92.89) | 91, 94 | 106 (7.11) | 6.0, 9.0 | <0.001 |
| 40 – 69 Years old | 11070(63.72) | 63.0, 64.0 | 10673(96.41) | 96, 97 | 397 (3.59) | 3.0, 4.0 | |
| ≥ 70 Years old | 4814 (27.71) | 27.0, 28.0 | 4708 (97.80) | 97, 98 | 106 (2.20) | 2.0, 3.0 | |
| Mean age | 60.18±14.93 | – | 60.36±14.88 | – | 55.20±15.22 | – | <0.001 |
| **Household Head Gender** | | | | | | | |
| Male | 8196 (47.17) | 46.0, 48.0 | 7927 (3.28) | 96, 97 | 269 (3.28) | 3.0, 4.0 | 0.131 |
| Female | 9178 (52.83) | 52.0, 54.0 | 8838 (96.30) | 95, 96 | 340 (3.70) | 3.0, 4.1 | |
| **Household head marital status** | | | | | | | |
| Single | 4141 (23.83) | 23.2, 24.5 | 4024 (97.17) | 96.62, 97.66 | 117 (2.83) | 2.34, 3.38 | <0.001 |
| Married | 5760 (33.15) | 32.5, 33.9 | 5596 (97.15) | 96.69, 97.57 | 164 (2.85) | 2.43, 3.31 | |
| Divorced | 7234 (41.64) | 40.9, 42.4 | 6924 (95.71) | 95.22, 96.17 | 310 (4.29) | 3.83, 4.78 | |
| Widowed | 197 (1.13) | 0.98, 1.30 | 181 (91.88) | 87.15, 95.29 | 16 (8.12) | 4.71, 12.85 | |
| **Household head's educational status** | | | | | | | |
| No formal education | 627 (8.83) | 8.0, 10.0 | 612 (97.61) | 96, 99 | 15 (2.39) | 1.40, 3.00 | 0.884 |
| Primary education | 1035 (14.57) | 14, 15 | 1006 (97.20) | 96, 98 | 29 (2.80) | 2.0, 4.0 | |
| Secondary education | 5253 (73.97) | 73, 75 | 5106 (97.18) | 97, 97 | 148 (2.82) | 2.0, 3.0 | |
| Tertiary education | 187 (2.63) | 2.0, 3.0 | 183 (97.86) | 94, 99 | 4 (2.14) | 1.0, 6.1 | |
| **Household head employment status** | | | | | | | |
| Employed | 1888 (26.64) | 26, 28 | 1836 (97.25) | 96, 98 | 52 (2.75) | 2.0, 4.0 | 0.993 |
| Not employed | 5199 (73.36) | 72, 74 | 5056 (97.25) | 97, 98 | 143 (2.75) | 2.0, 3.0 | |
| **Household crowding** | | | | | | | |
| Low crowding | 5876 (33.86) | 33, 35 | 5732 (97.55) | 97, 97 | 114 (1.45) | 2.0, 3.3 | <0.001 |
| Medium crowding | 6173 (35.57) | 35, 36 | 5965 (96.63) | 96, 96 | 208 (3.37) | 3.0, 4.0 | |
| High crowding | 5306 (30.57) | 30, 31 | 5049 (96.16) | 94, 96 | 257 (4.84) | 1.6, 5.6 | |
| **Wealth index quintiles** | | | | | | | |
| Poor | 5792 (33.34) | 33, 34 | 5578 (96.31) | 96, 97 | 214 (3.69) | 3.0, 4.0 | <0.001 |
| Middle | 5791 (33.33) | 33, 34 | 5633 (97.27) | 97, 98 | 158 (2.73) | 2.0, 3.0 | |
| Rich | 5791 (33.33) | 33, 34 | 5554 (95.91) | 95, 96 | 237 (4.09) | 4.0, 5.0 | |

and social grants are reported to offer a protective barrier against food insecurity [44–46]. This may be the case in the DIMAMO area, where indigenous subsistence crops and livestock production continue to be the primary sources of food and income in the area [47]. The low proportion of food insecurity indicates the significance of sustaining local subsistence farms, economic activities (livestock production), and strengthening the social grant system, as these improve the household food security [48]. These factors should therefore be considered when designing interventions and social protection policies for food security.

The present study found that the household head's gender, educational status, and employment status did not significantly associate with food insecurity. The lack of association may be explained by contextual factors within rural South African communities. In these settings, household food access is often supported by social protection mechanisms such as government grants, as well as subsistence farming and informal livelihood activities. These factors may buffer households from food shortages regardless of the household head's gender, education level, or formal employment status.

Contrary to the findings of the present study, Kara and Kithu [49] reported that as the educational attainment of the head of household increased, the level of household food security also improved [49]. The discrepancies between the present study and that of Kara and Kithu [49] may be as a result of multiple confounding factors. Education measures years of schooling or the highest qualification and may not capture economic returns, such that household heads with similar educational levels may have different economic attainments [50]. As a result, some household heads, although not educated, may be informally employed, with purchasing power needed to improve food security. Lastly, the inconsistencies between the two studies may be due to different methodological approaches and localities [48].

Relating to household gender, the findings of the present study were incongruent with the literature, as other studies have found associations between the household head's gender, educational status, and employment status. For instance, a study by Nagesse et [19], reported that female-headed households had 1.94 times the odds of developing food insecurity as compared with male-headed households. The findings were further confirmed by a systematic review that found that the odds of moderate and severe food insecurity were, respectively, 32% and 16% higher among households headed by women compared to households headed by men [51]. This discrepancy suggests that factors such as indigenous subsistence crops and livestock production reported in the DIMAMO area offer protection against food insecurity. Rural African communities, such as the DIMAMO area, are known to have social networks and community food-sharing practices that mitigate gender-related disparities observed elsewhere [52]. This indicates the significance of women's empowerment initiatives in policy making, even in areas where food insecurity appears to be relatively low [53,54].

In this study, the prevalence of food insecurity declined as age increased. Additionally, both bivariable and multivariable regression analyses indicated that an older age of the household head served as a protective factor against food insecurity. The food insecurity in young household heads could be linked to high unemployment, which is often more prevalent in younger individuals in the DIMAMO HDSS [55]. The household unemployment status may affect the type, quantity, and frequency of meals prepared at home. Contrary to the findings of the present study, previous studies noted that food insecurity is more prevalent among older individuals [56,57]. A study by Koyanagi et al. [58], reported that individuals aged above 65 years were 3.87 times more likely to experience food insecurity. Nevertheless, the findings of the present study are still justifiable. Rural areas such as the DIMAMO HDSS have a high number of elderly people who are recipients of the elderly grants. These grants assist in alleviating food insecurity [59–61]. This finding indicates the importance of maintaining and expanding the social grant system to ensure sustainability, while also creating employment opportunities for younger household heads to reduce their vulnerability to food insecurity [62].

Marital status also played a major role in food insecurity in the study. Widowed and divorced individuals were reported to have a higher proportion of food insecurity (8.12% and 4.29%, respectively) compared to those who are single or married. Bivariable logistic regression reported that married and widowed household heads were associated with food insecurity. However, on multivariable regression, after adjusting for confounders, only the association between the widowed household heads and food insecurity remained. Suggesting that the association between married household heads and food insecurity may have been modulated by other factors. In agreement with the findings of the present study, previous studies reported similar findings with widowed household heads being at risk of food insecurity compared to other marital statuses [63,64]. One possible reason for the above findings may be that the death of a spouse often results in the loss of a primary or secondary source of income. In many cases, especially among older widowed women, the deceased spouse may have been the primary breadwinner, leaving the surviving partner with limited financial resources. Targeted support interventions such, as food vouchers and community nutrition programs, could therefore be beneficial for widowed and divorced households who may lack adequate economic support systems [65,66].

The present study found that households in the middle category of wealth index were less likely to be food insecure. Household wealth index, determined using household assets and characteristics, a proxy for household economic well-being, has been linked to food insecurity, as it can influence the type of food the household has access to [67,68]. Household assets (e.g., tractors and livestock) offer an opportunity for additional income/ income diversity [69,70].

However, the present study household heads with a higher wealth index were at a higher risk of food insecurity. In line with the findings of the present study, research conducted in Ghana reported that household heads who had highly diversified incomes were more likely to report experiencing severe food insecurity [71]. Although households may have assets such as livestock, land, etc., the same may not translate to income, and thus negatively affect the household's food security [72]. This finding calls for policies that promote the productive utilization of household assets, such as improving market access, livestock commercialization, and financial literacy programs, to ensure that asset ownership effectively translates to food security gains [73]. From the results of the descriptive analysis, it can be noted that a higher proportion of food insecurity existed among households that fell into the rich category compared to those that fell into the poor category. This could be attributed to the fact that a wealth index is used instead of an income or liquidity measure. In a rural setting, households can have assets and still experience periods of food insecurity. Additionally, larger households and higher levels of consumption can create pressures on food expenditure. However, after controlling for confounding variables, it can be noted that higher levels of wealth were associated with reduced odds of food insecurity.

Household crowding played a significant role in food insecurity. Households with high crowding had the highest level of food insecurity. In addition, on both bivariable and multivariable regression analysis, households that were of crowding level medium and high were more likely to be food insecure. Household crowding conditions can increase the likelihood of food insecurity due to financial constraints. This is because in a large household, more food is required, thus increasing the burden on the income, more so if the income is not proportional to the household's needs. In agreement with the findings of the present study, previous studies reported similar findings, noting that larger household sizes were significantly associated with increased food insecurity, indicating that households with six or more members were at higher risk of being food insecure [9,18]. Therefore, household-level interventions should aim to improve resource management and promote family planning awareness to prevent overburdening household resources [74].

## Limitations and strengths of the study

The study was cross-sectional in nature the causal relationships between household/household head characteristics and food insecurity could not be established. In addition, the relationships between household/household head characteristics and food insecurity are to be interpreted as correlations rather than causations. Based on the self-reported data collected through household surveys can result in a recall bias. Although the use of self-reported 12-month recall data may introduce recall bias, this approach allowed the study to capture seasonal and chronic food insecurity patterns in a rural, rain-fed agricultural setting.

Another methodological issue is the relatively low level of food insecurity that was identified in the results of the current study (3.51%), which could effect the dependent variable's fluctuation and hence the regression results. However, the large sample size of the current study (n = 17,374) was adequate to detect significant relationships. Moreover, the diagnostic tests did not reveal any problematic multicollinearity in the regression models, implying that the regression models were stable despite the low level of food insecurity in the sample. The present study used secondary data. Hence, the study did not analyse for all possible determinants of food insecurity (i.e., dietary diversity, seasonal variations in food availability). The findings of the present study may not be generalized to other regions, since the study was conducted in the Capricorn region of Limpopo province. Additionally, the food insecurity measure was adapted from HFIAS and did not include all original questions, which may limit the scope of experiences captured. Nevertheless, the study provides valuable insights into the socio-demographic and structural factors influencing household food security in rural Limpopo. Despite the above limitations, the findings of the study highlight the importance of social protection mechanisms, such as social grant as they ensure sustainability and contribute towards food security in rural settings. The association between household crowding and food insecurity highlights the need for interventions that will focus on family planning, housing, and livelihood.

Furthermore, targeted social support and food assistance programmes should be provided for widowed household heads, given their increased vulnerability to food insecurity.

## Conclusion

The present study noted a low prevalence of food insecurity (3.51%). The low prevalence of food insecurity suggests that social protection mechanisms and local indigenous farming are a crucial barrier to households' food scarcity. The age of the household head was reported to be a protective factor for food insecurity, with food insecurity decreasing as age increased. The protective role of older age highlights the importance of income stability provided by old-age grants; hence, similar financial safety mechanisms could be extended to unemployed younger household heads. Households in higher wealth quintiles were less likely to be food insecure. Household crowding was reported to be associated with food insecurity, where higher household crowding was associated with higher vulnerability. There's a need for family planning and livelihood interventions that aim to reduce economic challenges in crowded households. Married and widowed household heads were also reported to be associated with food insecurity. There should be targeted interventions for widowed households, such as social support programs (food assistance and income-generating initiatives), to minimize food insecurity in this group with such a demographic. Future longitudinal studies should focus on the causal association between social protection, household composition, and food security to inform policy design at the disadvantaged local communities and provincial levels.

## Supporting information

**S1 Table. Crude and adjusted odds ratios.**
(DOCX)

**S1 Fig. Bivariate logistic regression of food insecurity status by household and household head characteristics.**
(DOCX)

**S2 Fig. Multivariate logistic regression of food insecurity status by household and household head.**
(DOCX)

## Acknowledgments

This research was carried out under the South African Population Research Infrastructure Network (SAPRIN), an initiative led by the South African Medical Research Council with sustained support from the National Department of Science and Innovation. We gratefully acknowledge this long-term funding, which has enabled DIMAMO to consistently collect data from the designated rural areas.

## Author contributions

**Conceptualization:** Reneilwe G. Mashaba, Kagiso P. Seakamela.

**Formal analysis:** Reneilwe G. Mashaba, Cairo Bruce Ntimana, Katlego Mothapo, Kurisani M. Mabhedle, Joseph Tlouyamma, Kagiso P. Seakamela.

**Methodology:** Reneilwe G. Mashaba, Cairo Bruce Ntimana, Katlego Mothapo, Kurisani M. Mabhedle, Joseph Tlouyamma, Kagiso P. Seakamela.

**Writing – original draft:** Reneilwe G. Mashaba, Cairo Bruce Ntimana, Katlego Mothapo, Kurisani M. Mabhedle, Joseph Tlouyamma, Kagiso P. Seakamela.

**Writing – review & editing:** Reneilwe G. Mashaba, Cairo Bruce Ntimana, Katlego Mothapo, Kurisani M. Mabhedle, Joseph Tlouyamma, Kagiso P. Seakamela.

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
