## [Decision Letter · Decision Letter 0]

22 Oct 2025

PGPH-D-25-01536

Food Security in Rural South Africa: The Role of Household Head Demographics, Crowding, and Wealth

Dear Dr. Ntimana,

Thank you for submitting your manuscript to PLOS Global Public Health. After careful consideration, we feel that it has merit but does not fully meet PLOS Global Public Health’s publication criteria as it currently stands. Therefore, we invite you to submit a revised version of the manuscript that addresses the points raised during the review process.

The manuscript has been evaluated by three reviewers, and their comments are available below.

The reviewers have raised a number of concerns that need attention. In particular, they request additional information on methodological and statistical aspects of the study, as well as improvements to the Discussion and overall presentation.

Could you please revise the manuscript to carefully address the concerns raised?

We look forward to receiving your revised manuscript.

Kind regards,

Helen Howard

Staff Editor

Journal Requirements:

1. In the online submission form, you indicated that “Data used in the manuscript will be provided upon reasonable request from the corresponding author.”

a. In a public repository,

b. Within the manuscript itself, or

c. Uploaded as supplementary information.

2. Please provide a/amend your detailed Financial Disclosure statement. This is published with the article. It must therefore be completed in full sentences and contain the exact wording you wish to be published.

a. If any authors received a salary from any of your funders, please state which authors and which funders.

Additional Editor Comments (if provided):

Reviewers' comments:

Reviewer's Responses to Questions

**Comments to the Author**

1. Does this manuscript meet PLOS Global Public Health’s publication criteria ? Is the manuscript technically sound, and do the data support the conclusions? The manuscript must describe methodologically and ethically rigorous research with conclusions that are appropriately drawn based on the data presented.

Reviewer #1: Yes

Reviewer #2: Yes

Reviewer #3: Partly

2. Has the statistical analysis been performed appropriately and rigorously?

Reviewer #1: Yes

Reviewer #2: Yes

Reviewer #3: I don't know

3. Have the authors made all data underlying the findings in their manuscript fully available (please refer to the Data Availability Statement at the start of the manuscript PDF file)?

Reviewer #1: No

Reviewer #2: Yes

Reviewer #3: Yes

4. Is the manuscript presented in an intelligible fashion and written in standard English?

Reviewer #1: Yes

Reviewer #2: Yes

Reviewer #3: No

Reviewer #1: Summary

In this paper, the authors clearly determined the relationship between household food security and the demographics of household heads as well as household level characteristics in rural South Africa.

The authors contributed to the existing knowledge on how variables such as age, marital status, crowding, employment status among other factors can influence food security of a population. The constructs were well defined, the data well analyzed and results clearly presented. The authors concluded that food insecurity decreases as age increases, making age a protective factor for food insecurity.

Overall, the research findings are interesting and of public health importance. It can be a handy tool to policy makers in creating a balance in food security among the populace.

Major Revision

The authors mentioned in the methodology that the study utilized secondary data collected from the Dikgale Mamabolo and Mothiba Health and Demographic Surveillance System (HDSS) now referred to as DIMAMOD HDSS. Also reiterated in line 66 “For this, 17,374 household heads were extracted from the DIMAMO PHRC database on 5th May 2025”. If this is correct, I am wondering why the authors wrote the ethical considerations as though they obtained the ethical approval and obtained written consent from all 17,374 participants. The authors should kindly clarify this important aspect of the work to avoid confusion.

Minor Revision

The authors presented all their results in tables which is very informative, I opine that graphical presentation will enhance interpretation by providing a clearer and more intuitive understating of the data.

Line 52. ……crowding and wealth index, in Limpopo province, South Africa (Preposition “in” omitted).

Line 142. The preposition “in” is equally missing.

Line 167: assist in elevating food insecurity [34–36]. (Should be alleviating).

Line 152-153: One study reported that as the educational attainment of the head of household increased, the level of household food security also improved [27]. Your result is not in tandem with this assertion, readers will be interested in possible reasons why? Authors should give robust explanation(s) to this and other results not in agreement with existing literature.

Line194-195: The present study noted a low prevalence of food insecurity (3.51%) in the DIMAMO HDSS, Limpopo Province, South Africa, which is low when compared to national and regional estimates. (Recast to improve clarity)

Reviewer #2: I want to commend you on your expertly crafted manuscript, which adheres impeccably to the PLOS Global Public Health guidelines. The findings you have presented are not only highly relevant but also pivotal to advancing research in the crucial areas of food security and nutrition. This study powerfully underscores the urgent need for similar assessments in various regions around the globe. Even a small percentage of individuals grappling with food insecurity and poor nutrition can profoundly undermine the socio-economic development of families, communities, and indeed, entire nations. Your work sheds light on this vital issue and calls for action that could have far-reaching benefits

Reviewer #3: COMMENTS

Article: Food Security in Rural South Africa: The Role of Household Head Demographics, Crowding, and Wealth

Thank you for the opportunity to review this paper.

Abstract

1. “This study aimed to investigate the association between household food security and the demographics of the household head, as well as household-level characteristics, in Limpopo Province, South Africa.” This sentence should be revised because, in its current form, it is unclear what the focal independent variables are. Additionally, it does not align well with the title.

2. “…used to describe the relationship between the binary dependent variable (Food secure = 0, Food insecure = 1) and the predictor variables.” What are the predictor variables?

3. Your cocnlusion is basically a repetition of the results.

Introduction

1. Food insecurity, defined as ‘insufficient accessibility or access to nutritious food or a limited/uncertain ability to buy food in socially acceptable ways is a persistent public health challenge [1,2].” Is your definition a direct quote? If so, note that the authors you are citing originally derived their definition from the American Dietetic Association. You should therefore reference it appropriately.

2. “In South Africa, the prevalence of food insecurity has been on the rise[6]. According to STATS SA, the proportion of households in South Africa that experienced moderate to severe food insecurity was estimated at 15,8% in 2019, 16,2% in 2022, and 19,7% in 2023 [7].” This reference seems to originate from a master’s thesis rather than the WHO as indicated, and it should be correctly cited.

3. “Furthermore, household and household head demographics have been reported to play a role in food security.” What does the literature say about this connection?

Methods

1. “The current study aimed to investigate the effect of the household head’s demographics on food security.” There needs to be consistency in this statement. How about wealth and crowding?

2. “To assess the relative socioeconomic status of households, a wealth index was constructed using the Principal Component Analysis (PCA).” How was this done? What were some of the variables used in deriving the variable?

3. Lines 73 – 74. I am curious why you chose to adapt 18 binary indicators from the HFIAS rather than using the original questions, given that the scale already includes 18 items. This approach is unclear and requires clarification. Additionally, is your scale valid and reliable?

4. Lines 81 – 83. Can you make reference to similar studies that use this criteria to measure crowding?

5. Please clarify how the binary food security variable was generated from the four original categories. The categories/classifications presented in Table 2 do not appear to align with those described in lines 77–79. It seems that you may have combined “marginal food security” with “high food security.” What is the rationale or theoretical justification for this decision?

6. “Variables with high multicollinearity were removed from the regression model.” Which variables?

7. Lines 96 – 100. I am curious is the authors used primary or secondary data. The detail information provided under this section suggests that the authors collected primary data.

8. Some typos exist in this se

Results

1. “Divorced household heads were associated with food insecurity (AOR=5.39; 95%CI:2.88-10.12; P<0.001).” What does AOR mean?

2. I recommend merging your tables. Do you really need table 2?

Discussion

1. The discussion lacks depth and primarily reiterates the results without a meaningful contextual analysis. What recoomendations do you have in terms of your findings?

2. The authors have not provided any limitations to the study.

3. Your conclusion once again is a repetition of your results. What are the implicatins of your findings?

4. There are numerous grammatical errors and typos throughout the paper, and it requires thorough proofreading to improve clarity and readability. For instance, “Household heard Gender” in tables 3 and 4.

References

1. There are some erros in the references. For example, reference 4 is not properly done. Please check all references.

(what does this mean? ). If published, this will include your full peer review and any attached files.). If published, this will include your full peer review and any attached files.

**Do you want your identity to be public for this peer review?** For information about this choice, including consent withdrawal, please see our Privacy Policy .

Reviewer #1: No

Reviewer #2: No

Reviewer #3: No

---

## [Decision Letter · Decision Letter 1]

22 Dec 2025

PGPH-D-25-01536R1

Food Security in Rural South Africa: The Role of Household Head Demographics, Crowding, and Wealth

Dear Dr. Ntimana,

Thank you for submitting your manuscript to PLOS Global Public Health. After careful consideration, we feel that it has merit but does not fully meet PLOS Global Public Health’s publication criteria as it currently stands. Therefore, we invite you to submit a revised version of the manuscript that addresses the points raised during the review process.

We look forward to receiving your revised manuscript.

Kind regards,

Damen Haile Mariam, MD, MPH, PhD

Academic Editor

Journal Requirements:

Additional Editor Comments:

Reviewer 1:

- Abstract

1. “This study aimed to investigate the association between household food security and the demographics of the household head, as well as household-level characteristics, in Limpopo Province, South Africa.” This sentence should be revised

because, in its current form, it is unclear what the focal independent variables are. Additionally, it does not align well with the title.

2. “…used to describe the relationship between the binary dependent variable (Food secure = 0, Food insecure = 1) and the predictor variables.” What are the predictor variables?

3. Your conclusion is basically a repetition of the results.

- Introduction

1. Food insecurity, defined as ‘insufficient accessibility or access to nutritious food or a limited/uncertain ability to buy food in socially acceptable ways is a persistent public health challenge [1,2].” Is your definition a direct quote? If so, note that

the authors you are citing originally derived their definition from the American Dietetic Association. You should therefore reference it appropriately.

2. “In South Africa, the prevalence of food insecurity has been on the rise[6]. According to STATS SA, the proportion of households in South Africa that experienced moderate to severe food insecurity was estimated at 15,8% in 2019, 16,2% in

2022, and 19,7% in 2023 [7].” This reference seems to originate from a master’s thesis rather than the WHO as indicated, and it should be correctly cited.

3. “Furthermore, household and household head demographics have been reported to play a role in food security.” What does the literature say about this connection?

- Methods

1. “The current study aimed to investigate the effect of the household head’s demographics on food security.” There needs to be consistency in this statement. How about wealth and crowding?

2. “To assess the relative socioeconomic status of households, a wealth index was constructed using the Principal Component Analysis (PCA).” How was this done? What were some of the variables used in deriving the variable?

3. Lines 73 – 74. I am curious why you chose to adapt 18 binary indicators from the HFIAS rather than using the original questions, given that the scale already includes 18 items. This approach is unclear and requires clarification. Additionally, is

your scale valid and reliable?

4. Lines 81 – 83. Can you make reference to similar studies that use these criteria to measure crowding?

5. Please clarify how the binary food security variable was generated from the four original categories. The categories/classifications presented in Table 2 do not appear to align with those described in lines 77–79. It seems that you may have

combined “marginal food security” with “high food security.” What is the rationale or theoretical justification for this decision?

6. “Variables with high multicollinearity were removed from the regression model.” Which variables?

7. Lines 96 – 100. I am curious is the authors used primary or secondary data. The detail information provided under this section suggests that the authors collected primary data.

8. Some typos exist in this section.

- Results

1. “Divorced household heads were associated with food insecurity (AOR=5.39; 95%CI:2.88-10.12; P<0.001).” What does AOR mean?

2. I recommend merging your tables. Do you really need table 2?

- Discussion

1. The discussion lacks depth and primarily reiterates the results without a meaningful contextual analysis. What recommendations do you have in terms of your findings?

2. The authors have not provided any limitations to the study.

3. Your conclusion once again is a repetition of your results. What are the implications of your findings?

4. There are numerous grammatical errors and typos throughout the paper, and it requires thorough proofreading to improve clarity and readability. For instance, “Household heard Gender” in tables 3 and 4.

- References

1. There are some errors in the references. For example, reference 4 is not properly done. Please check all references.

Reviewer 2:

- Abstract

• The Conclusion section of the abstract seems the mere repetition of results and requires revision. The conclusion section should state key findings as per the specific objectives of study and doable recommendations to the point.

• The terms bivariate and multivariate need to be replaced by bivariable and multivariable.

• Data are plural (Data was versus Data are).

• The sampling procedure and methods of data collection need to be stated.

• All predictors need to be supported with measure of association (AOR) and 95% confidence interval consistently or described simply qualitatively.

- Introduction

• Food insecurity definition should follow the standard definition of Food and Agriculture Organization (FAO) definition “food insecurity occurs when people lack regular access—because of financial, physical, or other resource constraints—to

enough safe and nutritious food for normal growth, development, and an active healthy life.”

• What is really not known about predictors of food insecurity across the globe? There is plenty of evidence on predictors of food insecurity. What gaps the present study wanted fill is not convincing?

- Methods

• The study setting should describe the agro-ecological characteristics including the pattern of rain fall as it relates to food production.

• How feasible and valid is using a 12-months recall Food Security questions.

• Is it scientifically sound to combine combining “food secure” with “mild food insecure”?

• How assumptions of PCA were checked?

- Results

• Title of a table and figure should be complete answering three basic questions (what, where, and when).

• I suggest table 2 to be substituted with appropriate figure

• I suggest tables 3 & 4 to be merged as crude analysis and adjusted analysis in one table using crude odds ratio and adjusted odds ratio respectively.

• For crude analysis use “COR” instead of “OR”.

• Line 134-135 “With regards to the wealth index, the household heads who were from the rich category were 0.50 times less likely to be associated with food insecurity (p<0.001) [AOR=0.50,95%CI: (0.34 – 0.72)] –the interpretation needs to

be revised as follows. “The odds of food insecurity were reduced by 50% among study participants from rich wealth category compared with those form “poor” wealth category.

• All other results need to be interpreted in similar fashion.

- Discussion

• The first paragraph of the discussion section should first state the main aim of the study and paraphrase the key findings as per the specific objectives of the study before commencing the discussion of each key finding.

• Limitations of the study need to be discussed at the end of discussion in one paragraph.

• Implications of results need to be discussed as well.

- Conclusion:

• This section needs to summarize the key finding as per the specific aims of the study qualitatively with put repeating the actual figure, for instance, need of mentioning the prevalence of food insecurity as 3.51%.

Reviewer 3:

- On page 8, line 143, insert the figure number (currently it looks like Fig. Table S1.

Reviewers' comments:

Reviewer's Responses to Questions

**Comments to the Author**

Reviewer #2: All comments have been addressed

Reviewer #4: (No Response)

publication criteria ? Is the manuscript technically sound, and do the data support the conclusions? The manuscript must describe methodologically and ethically rigorous research with conclusions that are appropriately drawn based on the data presented.

Reviewer #2: Yes

Reviewer #4: Yes

3. Has the statistical analysis been performed appropriately and rigorously?

Reviewer #2: Yes

Reviewer #4: Yes

4. Have the authors made all data underlying the findings in their manuscript fully available (please refer to the Data Availability Statement at the start of the manuscript PDF file)?

Reviewer #2: Yes

Reviewer #4: Yes

5. Is the manuscript presented in an intelligible fashion and written in standard English?

Reviewer #2: Yes

Reviewer #4: Yes

Reviewer #2: The author(s)i have answered all necessary questions raised by reviewers; however, on page 8, line 143, insert the figure number (currently it looks like Fig. Table S1.

Reviewer #4: 1) General Comments:

Abstract

• The Conclusion section of the abstract seems the mere repetition of results and requires revision. The conclusion section should state key findings as per the specific objectives of study and doable recommendations to the point

• The terms bivariate and multivariate need to be replaced by bivariable and multivariable

• Data are plural (Data was versus Data are)

• The sampling procedure and methods of data collection need to be stated

• All predictors need to be supported with measure of association(AOR) and 95% confidence interval consistently or described simply qualitatively

Introduction

• Food insecurity definition should follow the standard definition of Food and Agriculture Organization (FAO) definition “food insecurity occurs when people lack regular access—because of financial, physical, or other resource constraints—to enough safe and nutritious food for normal growth, development, and an active healthy life.”

• What is really not known about predictors of food insecurity across the globe? There is plenty of evidence on predictors of food insecurity. What gaps the present study wanted fill is not convincing?

Methods :

• The study setting should describe the agro-ecological characteristics including the pattern of rain fall as it relates to food production

• How feasible and valid is using a 12-months recall Food Security questions.

• Is it scientifically sound to combine combining “food secure” with “mild food insecure”?

• How assumptions of PCA were checked?

Results :

─ Title of a table and figure should be complete answering three basic questions; what, where, and when.

─ I suggest to substitute table 2 with appropriate figure

─ I suggest to merge table 3 &4 as crude analysis and adjusted analysis in one table using crude odds ratio and adjusted odds ratio respectively

─ For crude analysis use “COR” instead of “OR”.

─ Line 134-135 “With regards to the wealth index, the household heads who were from the rich category were 0.50 times less likely to be associated with food insecurity (p<0.001) [AOR=0.50,95%CI: (0.34 – 0.72)] –the interpretation needs to be revised as follows. “ The odds of food insecurity were reduced by 50% among study participants from rich wealth category compared with those form “poor” wealth category.

─ All other results need to be interpreted in similar fashion.

Discussion:

─ The first paragraph of the discussion section should first state the main aim of the study and paraphrase the key findings as per the specific objectives of the study before commencing the discussion of each key finding.

─ Limitations of the study need to discussed at the end of discussion in one paragraph

─ Implications of results need to be discussed as well

Conclusion:

• This section needs to summarize the key finding as per the specific aims of the study qualitatively with put repeating the actual figure ,for instance, need of mentioning the prevalence of food insecurity as 3.51%

(what does this mean? ). If published, this will include your full peer review and any attached files.). If published, this will include your full peer review and any attached files.

**Do you want your identity to be public for this peer review?** For information about this choice, including consent withdrawal, please see our Privacy Policy .

Reviewer #2: No

Reviewer #4: **Yes:** Gudina EgataGudina Egata

---

## [Decision Letter · Decision Letter 2]

4 Mar 2026

PGPH-D-25-01536R2

Food Security in Rural South Africa: The Role of Household Head Demographics, Crowding, and Wealth

Dear Dr. Ntimana,

Thank you for submitting your manuscript to PLOS Global Public Health. After careful consideration, we feel that it has merit but does not fully meet PLOS Global Public Health’s publication criteria as it currently stands. Therefore, we invite you to submit a revised version of the manuscript that addresses the points raised during the review process.

We look forward to receiving your revised manuscript.

Kind regards,

Damen Haile Mariam, MD, MPH, PhD

Academic Editor

**Journal Requirements:**

**Additional Editor Comments (if provided):**

Reviewer 1 -

- Introduction:

- In lines 71-72 the author(s) stated that @For example, the role of household head demographics characteristics, household crowding, and wealth indices in influencing food security within rural South African communities remains not fully

explored (This statement needs evidence and support; may be the authors can cite other studies or show what were the emphasis of previous food security studies in South Africa).

- Methods:

- In line 87, the authors indicated that they used DIMAMO PHRC database. What is this data base? How appropriate is it for the study? what are the methodologies of data collection in DIMAMO PHRC data base?

- The authors also need to indicate what variables are extracted from the data base, the timeline of the data base? What is the rational for extracting 17,374 household heads? What is the population size in the data base?

- Results:

- Line 144-145, ’There was a significant difference (p-value = <0.001) in food security by household wealth index (poor: 3.69%, middle: 2.73%, and rich: 4.09%)’ Why do the rich have higher food insecurity than the poor?

- Line 150-151, Household head’s gender, educational status, and employment status were not associated with food insecurity. (Is there any explanation for this?)

- Discussion:

- The fact that there is a low overall prevalence of food insecurity (3.51%) reduces the variation in the dependent variable. This will have effects in the model. How is this addressed in the paper?

Reviewers' comments:

Reviewer's Responses to Questions

**Comments to the Author**

Reviewer #5: (No Response)

publication criteria ? Is the manuscript technically sound, and do the data support the conclusions? The manuscript must describe methodologically and ethically rigorous research with conclusions that are appropriately drawn based on the data presented.

Reviewer #5: Yes

3. Has the statistical analysis been performed appropriately and rigorously?

Reviewer #5: Yes

4. Have the authors made all data underlying the findings in their manuscript fully available (please refer to the Data Availability Statement at the start of the manuscript PDF file)?

Reviewer #5: Yes

5. Is the manuscript presented in an intelligible fashion and written in standard English?

Reviewer #5: Yes

Reviewer #5: In lines 71-72 the author(s) stated that @For example, the role of household head demographics characteristics, household crowding, and wealth indices in influencing food security within rural South African communities remains not fully explored (This statement needs evidence and support; may be the authors can cite other studies or show what were the empahasis of previous food security studies in South Africa

In line 87, the authors indicated that they used DIMAMO PHRC database. What is this data base? How appropriate is it for the study?; what the are the methodologies of data collection in DIMAMO PHRC data base

The authors also need to indicate what variables are extracted from the data base, the time line of the data base? What is the rational for extracting 17,374 house hold heads? What is the population size in the data base?

Line 144-145, ’There was a significant difference (p-value = <0.001) in food security by

household wealth index (poor: 3.69%, middle: 2.73%, and rich: 4.09%)’ Why do the rich have higher food insecurity than the poor

Line 150-151, Household head’s gender, educational status, and employment status were not associated with food insecurity. (Is there any explanation for this?)

The fact that there is a low overall prevalence of food insecurity (3.51%) reduces the variation in the dependent variable. This will have effects in the model. How is this addressed in the paper?

(what does this mean? ). If published, this will include your full peer review and any attached files.). If published, this will include your full peer review and any attached files.

**Do you want your identity to be public for this peer review?** For information about this choice, including consent withdrawal, please see our Privacy Policy .

Reviewer #5: No

---

## [Editor Report · Decision Letter 3]

10 Mar 2026

Food Security in Rural South Africa: The Role of Household Head Demographics, Crowding, and Wealth

PGPH-D-25-01536R3

Dear Mr Ntimana,

We are pleased to inform you that your manuscript 'Food Security in Rural South Africa: The Role of Household Head Demographics, Crowding, and Wealth' has been provisionally accepted for publication in PLOS Global Public Health.

Best regards,

Damen Haile Mariam, MD, MPH, PhD

Academic Editor